# Comparing Different Sticky Traps to Monitor the Occurrence of *Philaenus spumarius* and *Neophilaenus campestris*, Vectors of *Xylella fastidiosa,* in Different Crops

**DOI:** 10.3390/insects14090777

**Published:** 2023-09-21

**Authors:** Crescenza Dongiovanni, Michele Di Carolo, Giulio Fumarola, Daniele Tauro, Biagio Tedone, Simona Ancona, Valentina Palmisano, Mauro Carrieri, Vincenzo Cavalieri

**Affiliations:** 1Centro di Ricerca, Formazione e Sperimentazione in Agricoltura “Basile Caramia” (CRSFA), Locorotondo, 70010 Bari, Italy; enzadongiovanni@crsfa.it (C.D.); micheledicarolo@crsfa.it (M.D.C.); giuliofumarola@crsfa.it (G.F.); danieletauro89@gmail.com (D.T.); biagiotedone2@gmail.com (B.T.); ancona.simona@gmail.com (S.A.); valentinapalmisano1997@gmail.com (V.P.); maurocarrieri@crsfa.it (M.C.); 2Istituto per la Protezione Sostenibile delle Piante, CNR, 70126 Bari, Italy

**Keywords:** monitoring, *Xylella fastidiosa*, insect vectors, Apulia, spittlebugs, Aphrophoridae

## Abstract

**Simple Summary:**

Spittlebugs are the main vectors of the quarantine bacterium *Xylella fastidiosa* (Wells et al.) in Europe. Knowledge of the occurrence and population density of spittlebugs in different *Xylella*-susceptible crops has therefore become crucial to implement preventive and containment vector control strategies in order to reduce the risk of new outbreaks and the spread of the bacterium in infected areas. So far, sweep nets have been the most widely used method to monitor spittlebugs in different vegetation compartments in Europe. In this study, the attractiveness of different sticky traps, compared to sweep nets, has been examined. The obtained results suggest that the two methods should be integrated to achieve an accurate estimation of the presence and abundance of spittlebugs throughout the whole season when adults are present in the crops.

**Abstract:**

Following the detection of the quarantine bacterium *Xylella fastidiosa* (Wells et al.) in the Apulia region (southern Italy) and the identification of spittlebugs as the main vector species that contributes to its epidemic spread, monitoring activities have been intensified in an attempt to implement vector control strategies. To date, sweep nets have been the most widely used sampling method to monitor adult spittlebug populations. Field experiments were carried out, during 2018 and 2019, to evaluate the effectiveness of sticky traps in capturing spittlebugs in different woody crops. The attractiveness of different traps was compared: four colored sticky traps (white, red, blue, and yellow), with the yellow sticky traps having three different background patterns (plain yellow, yellow with a black circle pattern, and yellow with a black line pattern). In addition, the efficiency of the yellow sticky traps was evaluated by placing the traps on the ground or hanging them from the canopies in orchards with different spittlebug population densities. Trap catches of *Philaenus spumarius* (Linnaeus) and *Neophilaenus campestris* (Fallén) (Hemiptera: Aphrophoridae) were compared with those collected using sweep nets. The two spittlebug species showed a similar response to the colored traps and were mainly attracted to the yellow sticky traps. Captures throughout the adult season indicated that an accurate estimation of the presence and abundance of spittlebugs can be obtained by integrating the two sampling methods. Moreover, sweep nets appeared to be more efficient in collecting adults soon after their emergence, while the use of sticky traps was more efficient in the rest of the adult season when the use of traps can significantly expedite vector monitoring programs.

## 1. Introduction

True spittlebugs (Hemiptera: Aphrophoridae) have recently received considerable attention in Europe as they are vectors of the quarantine phytopathogenic bacterium *Xylella fastidiosa (Xf)*—Wells [1,2,3]. The report of *Xf* outbreaks in olive groves in southern Italy (in the Apulia region) in 2013 resulted in mandatory EU surveillance programs to enhance the discovery of unreported entrenched foci, but also in Corsica and the Balearic Islands, as well as to early-detect newly established infections, such as those discovered in the Provence-Alpes-Côte d’Azur region (France), in the region of Madrid, in Tuscany and Latium (Italy), and in Portugal [3,4,5]. Thus, subspecies assignment and genetic characterization of the strain(s), coupled with the assessment of the host range and identification of the insect vector species, should be a priority in the research programs of all the countries that have to deal with outbreaks of *Xf*.

The relevance of insect vectors in the epidemiology (spread and persistence of bacterial infections) and the lack of curative applications for *Xf*-infected plants make the vectors the primary target of control strategies to counteract the impact of *Xf*-epidemics [6].

Sharpshooters (Hemiptera: Cicadellidae: Cicadellinae) and spittlebugs (Hemiptera: Aphrophoridae) are the two taxonomical groups involved in the transmission of *Xf* throughout the world. The former are predominant in the American *Xylella*-pathosystems, while the latter are prevalent in European agroecosystems. Other xylem-sap feeders, belonging to the Cercopoidea, Cicadoidea and Membracidae families, are potential vectors of *Xf* [7]. Numerous potential vectors have been identified throughout Europe following pest risk assessments [2]. However, taking into account the polyphagia, abundance and distribution, it has been found that the groups most likely to play a significant role in Europe are four spittlebugs: *Philaenus spumarius* (Linnaeus), *Neophilaenus campestris* (Fallén) *Neophilaenus lineatus* (L.) and *Aphrophora alni* (Fallén), the froghopper *Cercopis vulnerata* (Rossi) and the sharpshooter *Cicadella viridis* (Linnaeus) [7]. Furthermore, the sharpshooter *Graphocephala fennahi* (Young), which is of American origin, has been reported to have a wide geographic distribution, including temperate northern European regions, and its host range is also of particular relevance [8]. The American sharpshooter *Draeculacephala robinsoni* (Hamilton) has recently been found for the first time in Europe and in the Palearctic regions of France (Pyrenées-Orientales) and Spain (Catalonia) in large populations, thus indicating that the species already seems to be established in these regions and could spread to other areas, as the adults of this species are considered strong flyers [9].

Investigations and surveys on the European outbreaks have confirmed that spittlebugs are the main species involved in the foci identified in Italy, Corsica, the Balearic Islands, and mainland Spain [3,10,11,12]. Three different species: *P. spumarius*, *Philaenus italosignus* (Drosopoulos & Remane) and *N. campestris* have been proven to be competent vectors of the isolates associated with the epidemic in olives, from and to different susceptible hosts (olives, oleander, cherry, almond) [13,14].

Estimating the relative abundance and seasonal fluctuation of the Auchenorrhyncha adult population on olive trees has clearly shown that the *P. spumarius* spittlebug and *N. campestris* are the two most common vector species to have been captured on crops and in natural environments [13,14,15,16].

Several approaches were adopted in the past to monitor the occurrence and the population density of spittlebugs, to both study their biology and to identify the most vulnerable stage for their control [11,14,15,16,17].

Sweep nets are the most widely applied sampling method of *P. spumarius* adults [11,17,18,19,20,21] but, as reported by Purcell et al. [22], although this method is highly effective for the evaluation of insects on ground cover, it is less effective on canopies. Attempts to use vacuum samplers, such as the backpack motor D-Vac, have provided poor results in the collecting of vectors of *X. fastidiosa* in vineyards, and on almond and olive trees [23].

Other authors have reported the use of sticky traps, aerial suction traps, beat trays, and tanglefoot bands for the monitoring of different vectors, but they have all proved to be less effective than sweep nets [24,25,26,27,28,29].

The use of colored sticky traps is a quick and simple method to obtain information on the distribution and population density of several insect pests that affect crops of economic importance. Sticky traps are widely used for Hemiptera to monitor psyllids [30,31], whiteflies [32,33], and leafhoppers, the pests or vectors of several plant pathogens [34,35]. The use of sticky traps to monitor adult spittlebugs is still considered controversial, due to the inconsistent results that have been obtained in terms of efficiency [20], and a study conducted in Spain [23] showed that this method is only slightly effective in capturing meadow spittlebugs. Previous experiments [26] showed that yellow sticky traps were more effective in catching meadow spittlebugs than other colored (green, red, pink, blue or white) traps.

The complex biology and phenology of spittlebugs [17,36], as well as the fluctuations and dispersion of the adults amongst the different vegetative compartments of crops and in neighboring areas, require the use of different approaches to efficiently monitor their occurrence and population densities in areas where vector control strategies have to be implemented. So far, sweep nets are the most commonly used approach in European countries that have to tackle *Xf* outbreaks and epidemics [20]. Therefore, the main objective of our work has been to assess whether the use of sticky traps could represent an efficient tool (less expensive and time-consuming than sweep nets) to monitor spittlebugs (*P. spumarius* and *N. campestris*) on a large scale in olive orchards and on the main agricultural crops (cherry and almond) of the Apulia region (southern Italy), which is currently threatened by the *Xf* epidemic.

The use of traps could result in more extensive vector surveillance campaigns, in order to implement vector control strategies, either where the bacterium is established or in areas with a high risk of establishment.

After preliminary evaluations, the traps were further validated, under different field conditions, and compared with sweep nets.

## 2. Materials and Methods

Experiments were carried out, in several steps, in open fields in the region of Apulia. First, the effectiveness of sticky traps of different colors was compared in olive and almond orchards. A more extensive validation of the yellow sticky traps was then carried out. To this end, we tested yellow sticky traps with different black line patterns, provided by Russel IPM Ltd. (Flintshire, UK). The traps were located either on ground vegetation or on the canopies, and the captures were compared with those recovered through the use of sweep nets.

### 2.1. Evaluation of the Different Colored Sticky Traps in Olive Groves and Almond Orchards

Six different colored sticky traps (30.5 cm × 20.5 cm), made up of polyethylene vinyl acetate (PVC), laminated on both sides with long-acting transparent glue, were tested to assess the attractiveness of different colored sticky traps: white, red, blue, yellow, yellow with a circle-pattern and yellow with a line-pattern (Russel IPM).

Trials were set up in two olive groves (the first one with 20-year-old trees, and the second one with century-old trees) and in one almond orchard (40 years old), located in the *Xylella*-free area of the Apulia region (Latitude: 40.78° N; Longitude: 16.42° E). The landscape in this area is heterogeneous, and it is characterized by the presence of alternating plots of olive groves, vineyards, almond and cherry orchards, and uncultivated areas. The site was selected based on previous monitoring data, which showed high population densities, and because of the low-input management practices in the area (only minimum tillage applied in spring and no insecticide applications over the last ten years).

The traps were randomized, according to their color, in a Latin Square Design and hung on the southwest-facing side of the trees at 1.6–1.8 m above the ground. Forty traps/color/olive grove were positioned on the olive trees (monitored surface 1 ha) and 20 traps/color were placed in the almond orchard (monitored surface 0.5 ha). The distance between traps depended on the distance between the plants on the rows, being 2.5 m in the young olive groves, 7.5 m in the century-old olive grove, and 6 m in the almond orchard.

The traps were located and inspected twice a month in June and July. Adults captured in the traps were identified directly in the field. Unidentified insects were unglued from the traps, employing ethyl acetate 99.8%, transferred into 50 mL falcon tubes containing 70% ethanol, and subsequently stored at −20 °C until morphological identification. Insects were identified according to Biedermann & Niedringhaus [37], Della Giustina [38], Holzinger et al. [39] and Ribaut [40].

### 2.2. Evaluation of the Efficiency of the Different Yellow Sticky Traps in the Olive Groves, and in the Cherry and Almond Orchards

Three different kinds of yellow sticky traps (plain yellow sticky traps; yellow traps with a black circle pattern and yellow with a black line pattern), provided by Russel IPM (Appendix A), were tested in the experiments.

The experiments were carried out over a two-year period and set up in two locations; the first one was the same one that is described in Section 2.1, denoted as site A, while the second one was site B (latitude 40.87° N; longitude: 17.23° E), which included a consociated orchard with different fruit trees, prevalently almond, olive, and cherry trees, where the control of *P. spumarius* populations was applied both at the juvenile stage (weed control by means of soil tillage) and adult stage (applications of formulations based on acetamiprid and deltamethrin). Site B was surrounded by woodlots and arable land on the north side, and by typical affiliated crops (prevalently olive, cherry, and almond trees) on the other borders.

The traps were placed as described above, that is, on the southwest-facing side of the trees at a height of 1.6–1.8 m above the ground.

The traps positioned in 2018 included: 40 yellow traps/type in the olive groves (surface 1 ha) and 20 in the almond orchard (surface 0.5 ha). In 2019, the experiments were extended to site B, with 24 traps for each type used in the olive and almond orchards in site A and 36 in site B, where 12 traps were placed for each of the three crop species. The traps were rotated along each row in both years to avoid position effects and were randomized, according to the different kinds of yellow sticky traps, in a Latin Square Design.

During the first year, the traps were placed from June to October, while they were placed from May to early September in the second year. Surveys were conducted once a month in 2018 and twice a month in 2019. The specimens caught on traps belonging to the Auchenorrhyncha suborder were counted and identified following the same procedures previously reported in Section 2.1.

### 2.3. Monitoring the Vectors on Different Vegetation Compartments

The plain yellow sticky traps were placed in the olive groves, and in the cherry and almond orchards on the southwest-facing side of the trees, at a height of 20 cm above the ground vegetation and on the canopies at a height of 1.6–1.8 m above the ground. Experiments were set up on the same sites (site A and site B) as previously described in Section 2.2. Sixty traps were positioned at each height during the first year in each plot and the number was then reduced to 36 during the second year. The traps were visually inspected and replaced bi-weekly, from April to October, for two consecutive years. The data recorded in the cherry and almond orchards and in the olive groves were pooled and analyzed.

Insects were also collected, by means of sweep nets, on the same sampling dates. Ten sweeps per tree were performed on 20 olive, almond, and cherry trees, while four sweeps were made in 30 randomly distributed sampling units on the ground vegetation.

### 2.4. Testing the Traps in Sites Characterized by Different Spittlebug Density Populations

We selected sites characterized by different juvenile population densities to assess the effectiveness of the traps in relation to the spittlebug population density. The juvenile populations were estimated by surveying the selected sites from March to early May. Surveys were carried out using quadrat sampling (0.25 m^2^), as described by Bodino et al. [17]. The vegetation and soil surface inside the quadrat were visually inspected for the presence of spittlebug nymphs, which were identified at the species level and then counted. The sites were categorized, on the basis of these preliminary surveys, as being characterized by low, medium or high-density populations of *P. spumarius,* and by low and high population densities of *N. campestris*.

The presence and the occurrence of trapped adult spittlebugs were then monitored every two weeks in the selected sites from May to October using both yellow stick traps (20 for each plot) positioned at a height of 1.6–1.8 m above the ground and 10 sweeps/tree performed on 20 trees in each experimental field.

### 2.5. Data Analyses

The data on the spittlebugs and other Auchenorrhyncha species collected from different colored traps in the experimental scheme detailed in Section 2.1 and Section 2.2 were analyzed in separate groups using repeated MANOVA (PAST: Paleontological Statistics Software 4.03 [41]) measures. The model included the treatment (color), time of the year (date), and different observed crops (olive, almond or cherry). Each month’s data were analyzed separately. Whenever a significant MANOVA score was observed, it was followed by repeated ANOVA measurements for individual spittlebug species (*P. spumarius* or *N. campestris*) and other Aphrophoridae, and the ANOVA tests were performed with CoStat, version 6.204 (CoHort Software, Berkeley, CA, USA).

The ANOVA tests were also used to analyze the effects of the different heights on capturing *P. spumarius* and to compare the effects of the three yellow sticky traps on the attractiveness of spittlebugs and other Auchenorrhyncha species in the olive groves and in the almond and cherry orchards. The data were then tested for normality using the Kolmogorov–Smirnov test and for the homogeneity of variance using Levene’s test (F test). These preliminary evaluations showed that, according to the normality test, the data distribution did not deviate from the Gaussian and that, according to the F test, the populations had equal variances. For this reason, the data were not transformed. After the ANOVA tests, the means were separated by means of Tukey tests [42].

Student *t*-tests were conducted on normalized data to separately compare the number of spittlebugs (*P. spumarius* and *N. campestris*) captured in the sweep nets with those captured on the yellow sticky traps, and to compare sweeps on weeds and on canopies with the contents on the yellow sticky traps captured in orchards with different spittlebug population densities. Statistical significance was accepted for *p*-values < 0.05 for all the data.

## 3. Results

### 3.1. Evaluation of the Different Colored Sticky Traps in the Olive Groves and Almond Orchards

The different tested color traps (white, red, blue, yellow, yellow with a circle-pattern and yellow with a line-pattern), pooled from both the olive groves and almond orchards, showed a significantly different attractiveness, during both surveys, for the *P. spumarius* spittlebugs: (MANOVA, June: Wilks’ γ = 0.43, F = 41.2, df1 = 3, df2 = 96, *p* ≤ 0.00001; July: Wilks’ γ = 0.42, F = 33.96, df1 = 3, df2 = 76, *p* ≤ 0.00001) and *N. campestris* ones: (MANOVA, June: Wilks’ γ = 0.55, F = 25.17, df1 = 3, df2 = 96, *p* ≤ 0.00001; July: Wilks’ γ = 0.80, F = 6.08, df1 = 3, df2 = 76, *p* = 0.00092) (Figure 1).

When analyzed separately, the yellow-colored traps were found to be more attractive for *P. spumarius* in all the plots, in the young and old olive groves and in the almond orchards, than the other colored sticky traps. On the other hand, no difference was recorded between the different tested yellow sticky traps (plain, yellow with a circle pattern, and yellow with a line pattern) located in the olive groves. The yellow traps with a line pattern captured a significantly larger number of adults of *P. spumarius* adults during the month of July in the almond orchards (Table 1).

No differences were recorded for the different colored sticky traps used for the capture of *N. campestris* adults (Figure 1) in the secular olive grove, where the population was very low (Table 1). However, the captures were higher for the yellow sticky traps than for the other traps in the 20-year-old olive grove and in the almond orchard (Table 1).

### 3.2. Evaluation of the Efficiency of the Different Yellow Sticky Traps in the Olive Groves, and in the Cherry and Almond Orchards

#### 3.2.1. Olive Groves

Overall, no difference in attractiveness was observed over the two years for either spittlebug species among the different traps in any of the monitored sites (yellow, yellow with a circle pattern and yellow with a line pattern) located in the olive groves (*P. spumarius* in site A: MANOVA 2018: Wilks’ γ = 0.84, F = 2.04, df1 = 5, df2 = 54, *p* = 0.087490; MANOVA 2019: Wilks’ γ = 0.96, F = 0.30, df1 = 9, df2 = 62, *p* = 0.9726; site B: MANOVA 2019: Wilks’ γ = 0.54, F = 1.62, df1 = 8, df2 = 15, *p* = 0.2009; *N. campestris* in site A: MANOVA 2018: Wilks’ γ = 0.96, F = 0.40, df1 = 5, df2 = 54, *p* = 0.8452; MANOVA 2019: Wilks’ γ = 0.93, F = 0.71, df1 = 7, df2 = 63, *p* = 0.6583 and in site B: MANOVA 2019: Wilks’ γ = 0.81, F = 1.12, df1 = 4, df2 = 19, *p* = 0.376).

#### 3.2.2. Almond and Cherry Orchards

No difference among the tested yellow traps was recorded for *N. campestris* in either year for any of the sampling dates on almond and cherry (Almond trees: MANOVA 2018 site A: Wilks’ γ = 0.87, F = 1.61, df1 = 5, df2 = 54, *p* = 0.1724; MANOVA 2019 site A: Wilks’ γ = 0.85, F = 1.14, df1 = 9, df2 = 62, *p* = 0.3489; MANOVA 2019 in site B: Wilks’ γ = 0.86, F = 0.45, df1 = 6, df2 = 17, *p* = 0.8322; cherry trees: MANOVA 2019 in site A: Wilks’ γ = 0.86, F = 1.21, df1 = 8, df2 = 63, *p* = 0.306; MANOVA 2019 in site B: Wilks’ γ = 0.80, F = 0.90, df1 = 5, df2 = 18, *p* = 0.5003).

On the contrary, the captures of *P. spumarius* in some cases showed significant differences. Likewise, in 2018, in site A on almond (MANOVA: Wilks’ γ = 0.75, F = 3.586, df1 = 5, df2 = 54, *p* = 0.007139) limited the inspection carried out in late June (MANOVA: Wilks’ γ = 0.87, F = 4.41, df1 = 2, df2 = 57, *p* = 0.0166) and during 2019 in site B on cherry trees (MANOVA: Wilks’ γ = 0.40, F = 2.78, df1 = 8, df2 = 15, *p* = 0.0417) during the inspection carried out in early July (MANOVA: Wilks’ γ = 0.63, F = 3.96, df1 = 3, df2 = 20, *p* = 0.0229).

According to the ANOVA test, the yellow traps with a line pattern were more attractive than the other sticky traps for both crop species and at both sites (ANOVA 2018 in site A on almond: F = 4.75; df1 = 2, df2 = 57, *p* = 0.0144; ANOVA 2019 in site B on cherry: F = 95.633; df1 = 2, df2 = 15, *p* = 0.0024) (Figure 2 and Figure 3).

#### 3.2.3. Occurrence of Non-Target Insect Species on the Traps

Leafhoppers were the predominant non-target insects captured on the yellow sticky traps in all the experimental fields, over the whole inspection period, and in all the crops (Appendix A). Although the numbers varied according to the period, the most frequent recorded species were: *Anoplotettix putoni* (Ribaut), *Thamnotettix zelleri* (Kirschbaum), and *Synophropsis lauri* (Horvart), followed by three unidentified species belonging to the Issidae family.

As observed for spittlebugs, the efficiency of the three tested yellow sticky traps located in the olive groves to trap leafhoppers and planthoppers (Cicadellidae and Issidae) did not generally vary significantly (MANOVA 2019 in site A: Wilks’ γ = 0.77, F = 2.015, df1 = 9, df2 = 62, *p* = 0.05238; MANOVA 2019 in site B: Wilks’ γ = 0.71, F = 0.75, df1 = 8, df2 = 15, *p* = 0.6484), in the almond orchards (MANOVA 2018 in site A: Wilks’ γ = 0.95, F = 0.55, df1 = 5, df2 = 54, *p* = 0.7316; MANOVA 2019 in site B: Wilks’ γ = 0.72, F = 0.69, df1 = 8, df2 = 15, *p* = 0.6885) and on the cherry trees (MANOVA 2019 in site A: Wilks’ γ = 0.91, F = 0.67, df1 = 9, df2 = 62, *p* = 0.7257; MANOVA 2019 in site B: Wilks’ γ = 0.47, F = 2.08, df1 = 8, df2 = 15, *p* = 0.1055). However, differences in captures on the different yellow sticky traps were occasionally recorded in the olive groves during the inspections carried out in site A in July 2018 (MANOVA 2018: Wilks’ γ = 0.87, F = 4.45, df1 = 2, df2 = 57, *p* = 0.0160) and in late May 2019 on almond trees (MANOVA 2019: Wilks’ γ = 0.88, F = 2.947, df1 = 3, df2 = 68, *p* = 0.03894). When analyzed separately in ANOVA tests, the sticky traps with a line pattern captured more individuals than the other two types of yellow traps, in both the olive groves (ANOVA test 2018 in site A: F = 4.85; df1 = 2; df2 = 62, *p* = 0.013) and in the almond orchards (ANOVA test 2019 in site A: F = 4.25; df1 = 1, df2 = 54, *p* = 0.0202) (Table 2).

### 3.3. Monitoring Vectors in the Different Vegetation Compartments

The data recorded on the yellow sticky traps placed near the ground vegetation or on canopies in site A showed the highest capture/trap on the almond trees, regardless of the year and the inspection date. The only exception was the capture recorded in October 2019 when the density population of *P. spumarius* recorded on the cherry trees was similar to that on the almond trees (Table 3).

Instead, no differences in the average number of specimens/traps were observed among the crops from May to June in site B. However, a variation in the population density was recorded in the following months, with larger populations on olive trees than on the other plant species. In October, the largest number of captures was recorded in the cherry orchard, and this was followed by the almond orchard and the olive groves (Table 3).

The number of captures on the traps placed at the ground level or hung on the tree canopies varied throughout the seasons. Significantly higher captures were recorded on traps placed near the ground vegetation in the first period of adult emergence (ANOVA: May: F = 26.59; df1 = 1, df2 = 118, *p* < 0.00001; June: F = 13.06; df = 1, *p* = 0.00004) and in September–October, when spittlebugs return to herbaceous ground vegetation (ANOVA: September site A 2018: F = 6.08; df1 = 1, df2 = 118, *p* = 0.0146; October site A 2018: F = 69.62; df1 = 1, df2 = 118, *p* < 0.00001; October site B 2019: F = 13.88; df1 = 1, df2 = 70, *p* = 0.0005). However, in August, most of the individuals were caught in traps placed on the tree canopies (ANOVA: site A 2018: F = 175.40; df1 = 1, df2 = 118, *p* < 0.00001; site A 2019: F = 6.79; df = 1, df2 = 70, *p* = 0.01; site B 2019: F = 6.01; df = 1, df2 = 70, *p* = 0.0174) (Figure 4 and Figure 5).

When the data collected from the traps were compared with those obtained from the sweep nets, differences were recorded in relation to the crops and to the season. Soon after emergence (approx. in early May), *P. spumarius* adults were prevalently caught by means of the sweep nets (Figure 6 and Figure 7), with significant statistical values in (site A 2019, on olive trees in the second part of May: t = 2.65; df = 19; *p* = 0.0058; site B 2019, on olive trees in the first part of June: t = 2.24; df = 19; *p* = 0.0153; site A 2019, on cherry trees in the first part of May (t = 2.18; df = 19; *p* = 0.0018), the second part of May (t = 5.87; df = 19; *p* < 0.00001) and the first part of June (t = 1.76; df = 19; *p* = 0.0433) (Appendix A). Conversely, during summer, the captures on the traps were higher than those from the sweeps (Appendix A; Figure 6 and Figure 7), except in the cherry orchards where the sweep nets appeared to be more efficient than the traps (in site A in the second part of June: t = 2.42; df = 19; *p* = 0.01; first part of September: t = 1.89; df = 19; *p* = 0.033).

The traps showed consistently higher captures than the sweeps for the almond trees in both sites (A and B) and in both years (2018 and 2019) (Appendix A; Figure 6 and Figure 7).

At the end of summer, when the adults returned to the ground vegetation, the *P. spumarius* adults were captured more efficiently on traps, particularly in the olive groves: site A in the first part of October 2018 (t = 3.09; df = 19; *p* = 0.0019), in the second part of October 2019 (t = 1.96; df = 19; *p* = 0.0284), in site B in the second part of October 2019 (t = 2.39; df = 19; *p* = 0.0107), as well as on the cherry trees in site B in the first part of October 2019 (t = 2.98; df = 19; *p* = 0.0024) and in the second part of October 2019 (t = 2.08; df = 19; *p* = 0.0218).

No difference was noted in the cherry tree orchards in site A between the two approaches at the end of September—the start of October, that is, when adults move from the canopy of the trees to the newly emerged weeds (Figure 6 and Figure 7).

### 3.4. Testing the Traps in the Sites Characterized by Different Spittlebug Density Populations

Since the population density of *P. spumarius* ranged from 0.75 ± 0.26 to 21.8 ± 2.85 nymphs/0.25 m^2^, the 9 groves were categorized as follows: “with a low population density” for recorded average values of 1.3 ± 0.4 nymphs/0.25 m^2^; “with a medium population density” for average values of 8.6 ± 1.77 nymphs/0.25 m^2^, and “with a high population density” for average values of 15.9 ± 2.45 nymphs/0.25 m^2^.

In the 6 experimental fields in which the presence of *N. campestris* juveniles was recorded, the number of specimens per unit ranged from 0.01 ± 0.00 to 29.4 ± 4.73 nymphs/0.25 m^2^. These 6 fields were then grouped into two categories: “with a low population density” for average values of 0.13 ± 0.087 nymphs/0.25 m^2^ and “with a high population density” for average values of 17.5 ± 4.4 nymphs/0.25 m^2^.

The data collected soon after the emergence of the adults were consistent with those reported in the previous paragraph, i.e., the sweep nets were more efficient than the sticky traps in capturing insects, both on weeds [1st part of May for low density (t = 2.55; df = 56; *p* = 0.005871), medium density (t = 3.25; df = 56; *p* = 0.000716) and high-density population (t = 5.59; df = 56; *p* < 0.00001) conditions; 2nd part of May (low density: t = 4.25; df = 56; *p* = 0.00002; medium-density: t = 6.50; df = 56; *p* < 0.00001; high-density: t = 8.30; df = 56; *p* < 0.00001)] and on canopies [1st part of May in orchards with a high-density population (t = 2.63; df = 56; *p* = 0.004846); in all the orchards in the 2nd part of May (low-density: t = 1.78; df = 56; *p* = 0.039178; medium-density: t = 2.24; df = 56; *p* = 0.013242; high-density: t = 4.30; df = 56; *p* = 0.000019)] (Figure 8).

The two methods performed equally well in the groves with low a population density, except for in the 2nd part of June (t = 2.36; df = 56; *p* = 0.010015), when the *P. spumarius* population, in general, reached its maximum peak and the traps caught significantly more adults than the sweep nets (Figure 8).

The traps were more efficient than the sweep nets in the olive groves with a medium-density population, in all the surveys, from the second part of June (t = 7.49, df = 56; *p* < 0.00001) to the last part of October (t = 6.09; df = 56; *p* < 0.00001).

No difference was recorded between the two methods from the second part of June to the last part of July in the olive groves with a high population density. However, the mean number of *P. spumarius* adults captured by traps was consistently higher than the number captured by sweep nets from August (t = 4.96; df = 56; *p* < 0.00001) to October (t = 6.27; df = 56; *p* < 0.00001).

In a similar way to *P. spumarius*, the *N. campestris* adults were mainly captured by means of sweep nets soon after the emergence of the adults (May or the first part of June), in all 6 olive groves, both on ground vegetation (low-density population: 2nd part of May: t = 2.72; df = 56; *p* = 0.00365; high-density population: 1st part of May: t = 4.07; df = 56; *p* = 0.00004; 2nd part of May: t = 7.13; df = 56; *p* < 0.00001) and on canopies (low-density population: 2nd part of May: t = 1.65; df = 56; *p* = 0.049877; high-density population: 1st part of May: t = 1.01; df = 56; *p* = 0.15573; 2nd part of May: t = 3.37; df = 56; *p* < 0.00049). Then, after a peak was recorded on the herbaceous cover in the second part of June, no individuals were captured during the summer and in autumn in the orchards with a low-density population of *N. campestris* using either the traps or the sweep nets, and no significant differences (t-student tests) were recorded between the two applied methods. However, adults were captured in the remaining 3 olive groves (high-density population of *N. campestris*), albeit in low numbers, with an average of 0.03–1.15 on the traps and of 0.03 to 0.34 in the sweep nets, thus confirming a significant difference between the two methods (t = 2.83; df = 56; *p* = 0.00265) (Figure 8).

## 4. Discussion

Visual cues, in combination with olfactory stimuli, play an important role in the host location and selection behavior of leafhoppers and planthoppers [43,44,45,46,47,48]. Although investigations on the behavioral responses of spittlebugs and the related taxa to colors are scarce [49,50], yellow sticky traps have already been used to capture Aphrophoridae [23,49,50,51,52,53,54,55,56,57,58]. In our study, we assessed the efficiency of sticky traps to monitor spittlebugs throughout the whole season in which adults are present on the crops and can act as *Xf* vectors in those areas where the bacterium is established.

We have shown that yellow sticky traps are the most attractive for *P. spumarius* in olive groves, and on almond and cherry trees, which may be hosts of different strains and subspecies of *Xf* [2,3,4].

The two target spittlebugs, *P. spumarius* and *N. campestris*, showed a similar response to the colored traps, with both species being attracted more by the yellow sticky traps than the white, red or blue traps. These results are in agreement with previous results pertaining to cereal pastures obtained by Wilson and Shade [26] for *P. spumarius*, and with a recent study carried out in the Balearic Islands [58]. However, López–Mercadal et al. [58] did not find any differences between yellow and red traps for *N. campestris*, although they found both were more effective than white or blue traps.

The use of different yellow traps (plain, yellow with a circle pattern and yellow with a line pattern) did not show any significant differences; even so, those with a line pattern were, to a limited extent, more efficient but at the same time less selective, as they attracted large populations of leafhoppers, thus requiring more efforts and well-trained personnel for the inspections.

When the sweep nets and sticky traps on the tree canopies were compared, the traps were found to be better at capturing adults from the tree canopies during summer under our experimental conditions. This trend was observed in the monitored olive and stone fruit species and under all conditions for low, medium, and high adult population densities. The results described here are consistent with the results of the study of Purcell et al. [22]. Similar behavior has also been observed in a recent study by Beal et al. [59] in California vineyards, with more adults being caught by sweep nets in May and June, and more adults being caught on traps between July to November.

The captures recorded on sticky traps on olive canopies in the summer during our surveys indicated that most likely in the past when only sweep nets were used, the populations were underestimated under similar conditions [14,17,20,36]; indeed, it should be pointed out that the efficacy of sweep nets depends to a great extent on the sampling design and on the operator’s attitude to sweep vegetation. Furthermore, this method only captures the insects that are present on the canopies at the time of the sampling, while sticky traps detect the real situation of the population present in a specific site throughout the entire monitoring period. The efficiency of sweep nets is also influenced by the vegetative and structural features of the canopy; in our study, a low efficiency was recorded on almond canopies, which are characterized by a low foliage density and rigid branches, while a better efficiency was recorded for cherry canopies, which are characterized by abundant foliage and more flexible branches.

Moving from the ground vegetation to the tree canopy, it was found that yellow sticky traps were consistently more efficient than sweep nets, regardless of the population density. Even in fields with low population densities, traps provided good evidence of the presence and abundance of the target species, thus making them an efficient alternative to sweep nets.

Our results appear to be different from those recorded by Morente et al. in Spain [23]. This difference can most likely be explained by considering: (1) the different population densities of *P. spumarius*, which are very low in Spain compared to our study areas; (2) the different number of traps placed in the two studies; six traps on three plants/field in their experiment, compared to 20 traps/field in our study; (3) the different climatic conditions and management systems of the two locations.

The *P. spumarius* captures on the traps were also influenced by their position in the vegetation compartment and by the seasonal migration of the spittlebugs.

The largest number of captures on the traps located near the ground vegetation were recorded in spring, when the adults move from the ground vegetation to the canopies, and then in autumn when the adults return to weeds to lay eggs. Thus, positioning traps on ground vegetation in spring may help to identify the best period for insecticide applications against *P. spumarius* adults, i.e., before they acquire and spread the bacterium [7,17]. As far as the traps located on the canopies (ca. 1.7 m above the ground level) are concerned, the largest number of individuals was captured in August, regardless of the population densities of the specific monitored sites.

No difference was observed between the two sampling methods (traps versus sweep nets) in the remaining periods, probably because adults move continuously from olive, almond, and cherry canopies, to shrubs or other plants in their search for suitable shoots for sustenance and/or for plants as shelter (high or low temperatures, and adequate humidity conditions) [17,23,36].

No substantial difference was observed between the two adopted sampling methods regarding *N. campestris*. A peak of adults was recorded for both methods soon after adult emergence on the ground vegetation and also in early summer on tree canopies in locations with a high-density population. The population disappeared from the crops soon after the peak of captures on the canopies, and, as also pointed out by several authors [14,20,36], moved on to shrubs, *Cupressaceae*, oak or other non-cultivated plants around the cultivated plot, in the search for suitable shelter during the dry season.

These results corroborate the negligible role that *N. campestris* may have played in the spread of the bacterium in the epidemic in the Apulia region: (1) because of the low frequency and population density recorded on the canopies of the main crops, i.e., low transmission events on non-preferred hosts [60]; (2) because of the lower *Xf*-transmission efficiency than *P. spumarius,* as demonstrated on olive, almond, cherry, periwinkle and myrtle-leaf milkwort [14], and (3) because of the low frequency of the *Xf*-positive specimens detected in the *Xf*-infected areas in Apulia [14]. However, further investigations are needed to fully understand the risk posed by this spittlebug in other habitats, even though other authors [19,23,61,62] reached similar results, that is, the complete disappearance of *N. campestris* during the summer on commercial orchards and its dispersion on other alternative host plants.

The present work contributes to supporting the implementation of vector monitoring strategies by providing experimental evidence on the effectiveness of sticky traps as a useful tool to facilitate spittlebug monitoring on different crops.

As suggested by Purcell et al. [22], a combination of two or more sampling methods is needed to provide an accurate estimation of the presence and density of insects, and seasonal fluctuations. Our results suggest that sweep nets and sticky traps should be combined during the season for an accurate estimation of spittlebug populations on crops.

Moreover, on the basis of the achieved results, it seems that sweep nets should be used soon after the emergence of adults. This remains the best approach to identify the most appropriate time for the first application in an infected area, where mandatory insecticide applications are foreseen, to reduce the rate of bacterial acquisition from infected canopies [17]. Positioning sticky traps on canopies later on in the season may be useful to monitor the populations and to evaluate the effectiveness of the applied insecticide applications. The use of sticky traps, which reduces the necessary investments, in terms of costs and needed resources, promotes the intensification of vector monitoring campaigns. This is crucial in the demarcated areas where control measures are mandatory and must be adapted to the local situations. In conclusion, vector monitoring campaigns should be adopted to support prevention measures in areas with a high risk of *Xf* establishment.

## Figures and Tables

**Figure 1 insects-14-00777-f001:**
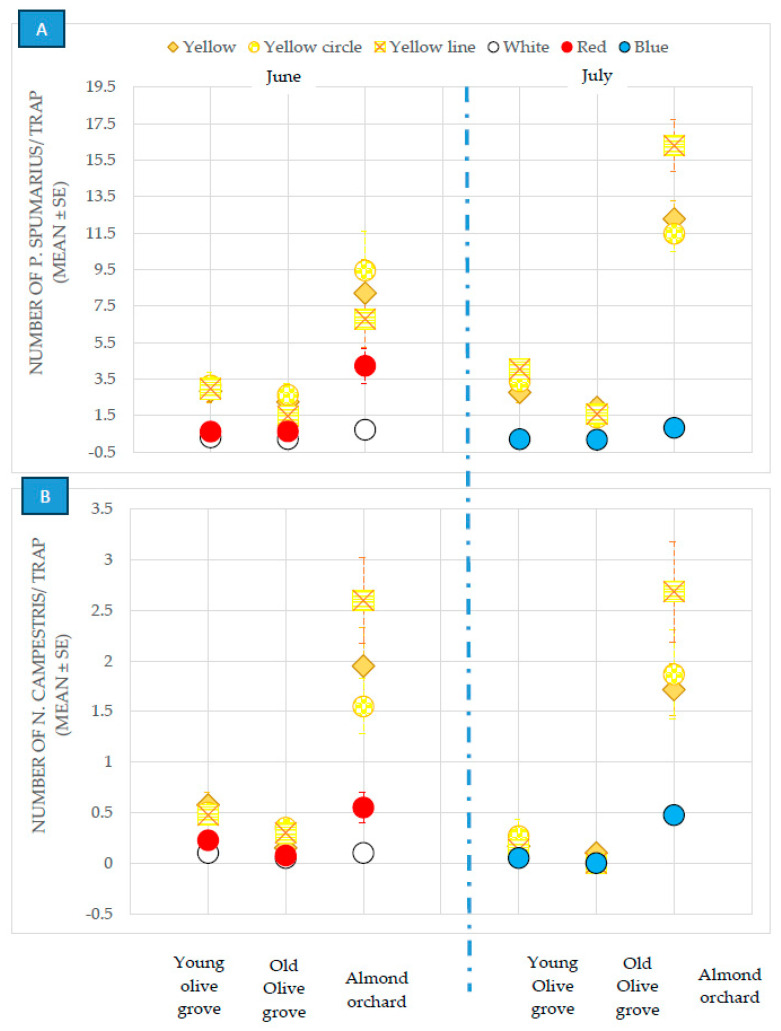
Captures of *Philaenus spumarius* (**A**) and *Neophilaenus campestris* (**B**) on sticky traps of different colors in the olive groves and almond orchards in 2018.

**Figure 2 insects-14-00777-f002:**
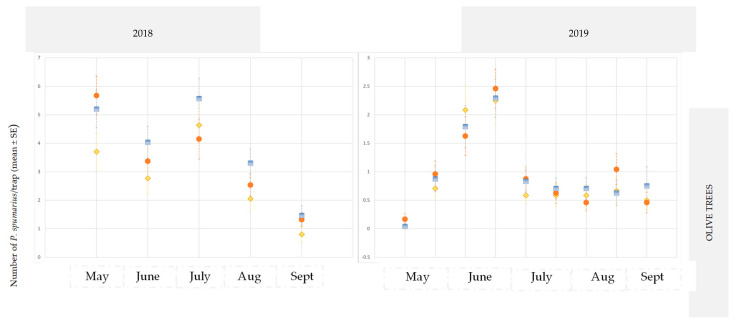
Captures of *Philaenus spumarius* adults on yellow sticky traps in the olive groves, and in the almond and cherry orchards in site A during 2018 and 2019. *** Significant at *p* < 0.0001.

**Figure 3 insects-14-00777-f003:**
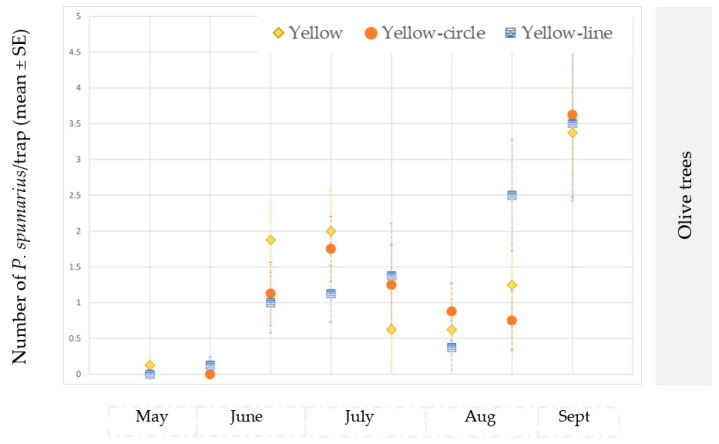
Captures on yellow sticky traps of adults of *Philaenus spumarius* in the olive groves, and in the almond and cherry orchards in site B during 2019. *** Significant at *p* < 0.0001.

**Figure 4 insects-14-00777-f004:**
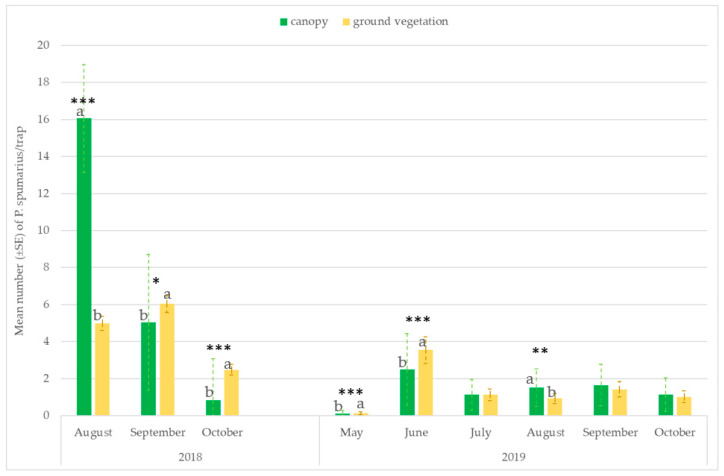
Mean number (±SE) of *Philaenus spumarius*/trap on traps positioned at two different heights: 20 cm above the ground vegetation and on the tree canopies (1.6–1.8 m). Pooled data of the olive groves, and the almond and cherry orchards in site A in 2018 and 2019. Values with the same letter are not significantly different. * Significant at *p* < 0.01; ** Significant at *p* < 0.001; *** Significant at *p* < 0.0001.

**Figure 5 insects-14-00777-f005:**
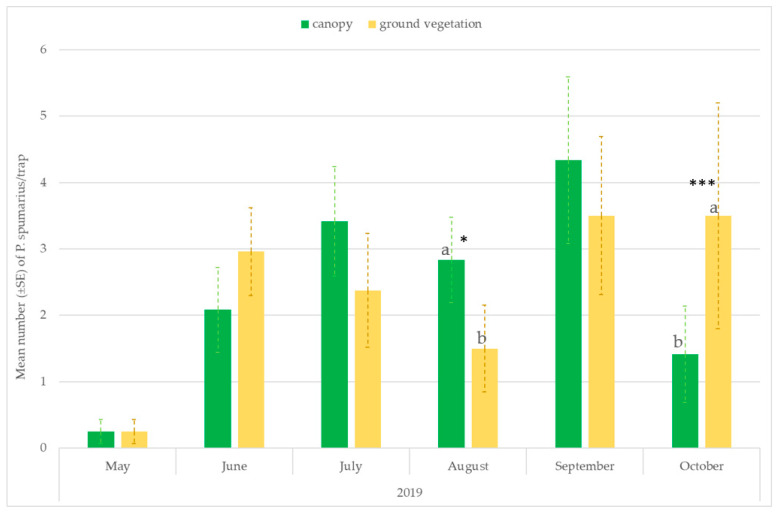
Mean number (±SE) of *Philaenus spumarius*/trap on traps positioned at two different heights: 20 cm above the ground vegetation and on the tree canopies (1.6–1.8 m). Pooled data of the olive groves, and the almond and cherry orchards in site B in 2019. Values with the same letter are not significantly different. * Significant at *p* < 0.01; *** Significant at *p* < 0.0001.

**Figure 6 insects-14-00777-f006:**
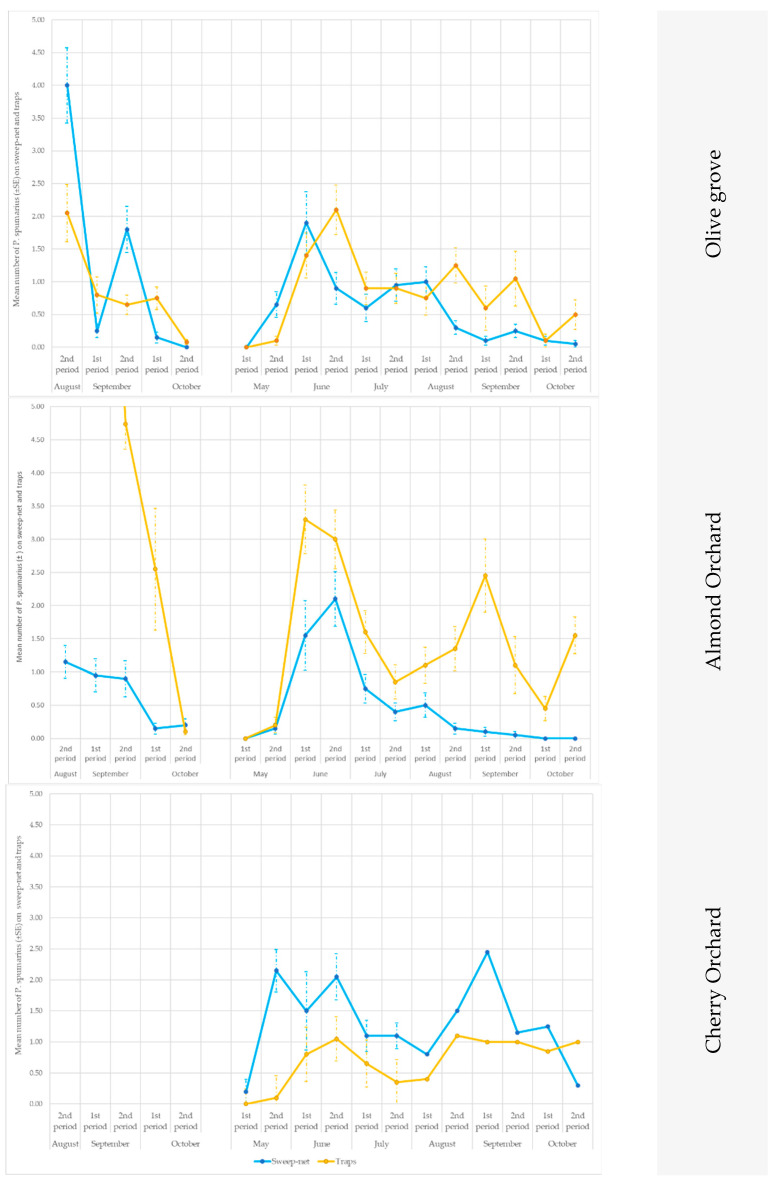
Seasonal pattern of the capture of *Philaenus spumarius* adults in the sweep nets and on the yellow sticky traps in the olive groves, and in the almond and cherry orchards in Site A during 2018 and 2019.

**Figure 7 insects-14-00777-f007:**
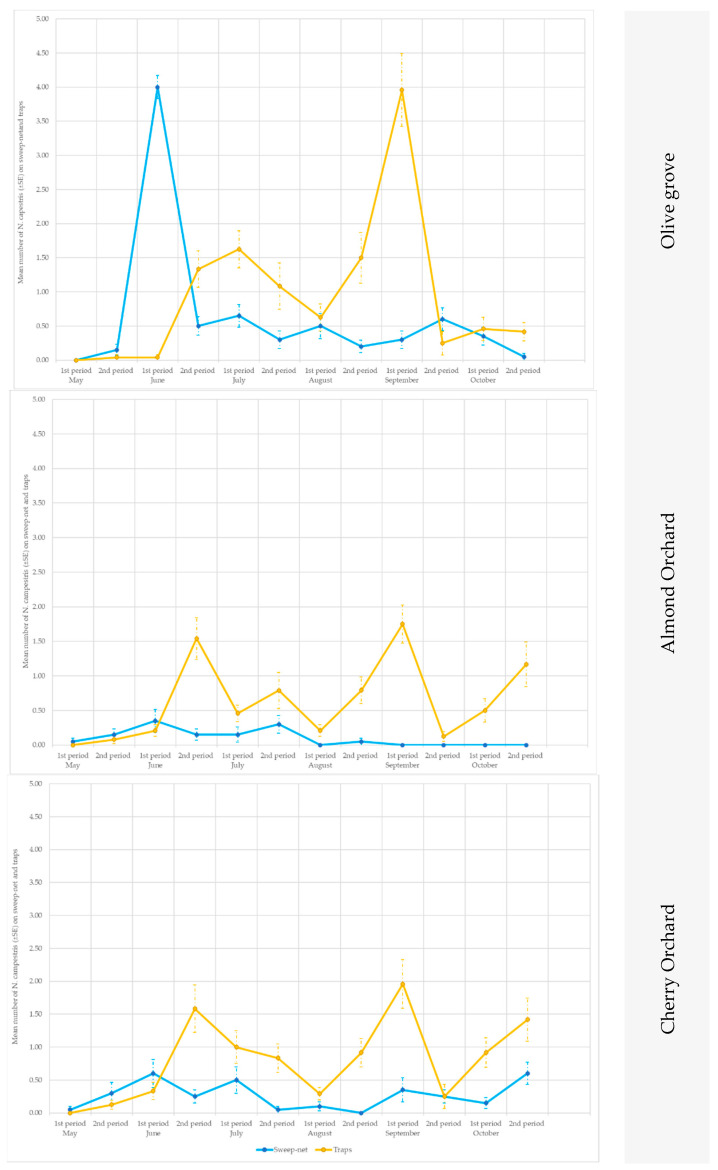
Seasonal pattern of the capture of *Philaenus spumarius* adults in the sweep nets and on the yellow sticky traps in the olive groves, and in the almond and cherry orchards in Site B during 2019.

**Figure 8 insects-14-00777-f008:**
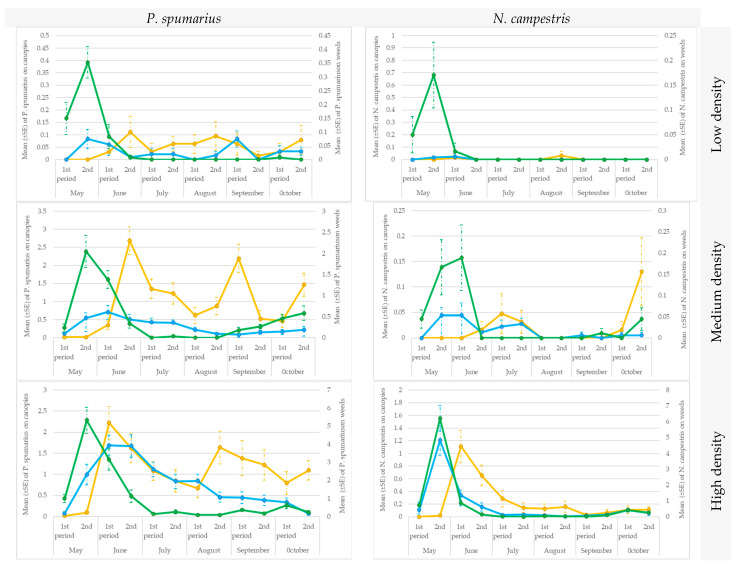
Seasonal pattern of the capture of *Philaenus spumarius* and *Neophilaenus campestris* adults under low, medium, and high-density spittlebug population conditions in Sites A and B during 2019. Ground cover with sweep nets (green line) and on canopies with sweep nets (blue line) or yellow sticky traps (yellow line).

**Table 1 insects-14-00777-t001:** Repeated ANOVA measurements pertaining to the effects of the trap color on the attraction of spittlebugs (*P. spumarius* and *N. campestris*) during inspections carried out in 2018. Values with the same letter are not significantly different.

**Traps**	**Mean Number of Spittlebugs (±SE)**					
	**June 2018**					
	** *P. spumarius* **			** *N. campestris* **		
	**20 Year-Old** **Olive Grove**	**Century-Old** **Olive Grove**	**Almond** **Orchard**	**20 Year-Old** **Olive Grove**	**Century-Old** **Olive Grove**	**Almond** **Orchard**
White	0.2 ± 0.06 b	0.1 ± 0.04 b	1.1 ± 0.16 b	0.1 ± 0.05 b	0.05 ± 0.03 a	0.2 ± 0.06 c
Red	0.75 ± 0.13 b	0.6 ± 0.14 b	3.4 ± 0.94 b	0.3 ± 0.09 ab	0.08 ± 0.04 a	0.7 ± 0.15 bc
Yellow	3.7 ± 0.63 a	2.6 ± 0.50 a	13 ± 1.83 a	0.6 ± 0.12 a	0.15 ± 0.06 a	2.2 ± 0.38 a
Yellow with a circle-pattern	5.7 ± 0.72 a	3.3 ± 0.60 a	14.3 ± 2.16 a	0.5 ± 0.10 a	0.35 ± 0.09 a	1.6 ± 0.27 ab
Yellow with a line-pattern	5.2 ± 0.68 a	2.3 ± 0.33 a	14.8 ± 1.55 a	0.5 ± 0.09 a	0.30 ± 0.08 a	2.6 ± 0.42 a
F-value	31.27	17.14	19.02	5.46	1.22	15.2
*p*-value	<0.00001	<0.00001	<0.00001	0.0006	0.2500	<0.00001
**Traps**	**July 2018**					
	** *P. spumarius* **			** *N. campestris* **		
	**20 Year-Old** **Olive Grove**	**Century-Old** **Olive Grove**	**Almond** **Orchard**	**20 Year-Old** **Olive Grove**	**Century-Old** **Olive Grove**	**Almond** **Orchard**
Blue	0.2 ± 0.05 b	0.2 ± 0.06 b	0.8 ± 0.12 c	0.05 ± 0.03 a	0 ± 0 a	0.5 ± 0.10 b
Yellow	2.8 ± 0.54 a	1.9 ± 0.43 a	12.3 ± 0.96 ab	0.2 ± 0.08 a	0.1 ± 0.06 a	1.8 ± 0.26 ab
Yellow with a circle-pattern	3.4 ± 0.69 a	1.4 ± 0.29 ab	11.5 ± 0.96 b	0.3 ± 0.16 a	0 ± 0 a	1.9 ± 0.44 ab
Yellow with a line-pattern	4.0 ± 0.56 a	1.6 ± 0.29 ab	16.3 ± 1.40 a	0.1 ± 0.07 a	0 ± 0 a	2.8 ± 0.50 a
F-value	10.58	5.74	40.84	0.33	2.11	5.09
*p*-value	<0.00001	0.0017	<0.00001	0.8008	0.1089	0.0034

**Table 2 insects-14-00777-t002:** Repeated ANOVA measurements for the effects of the trap color on the capture of other Auchenorrhyncha species in the olive groves and almond orchard in site A. Values with the same letter are not significantly different.

**Traps**	**26 July 2018**		
	**20 Year-Old Olive Grove**	**Century-Old Olive Grove**	**Almond Orchard**
Yellow	20.59 b	7.90 a	10.23 a
Yellow with a circle-pattern	27.53 ab	4.93 a	4.40 a
Yellow with a line-pattern	35.96 a	5.72 a	7.36 a
F-value	4.85	1.81	0.99
*p*-value	0.013	0.1779	0.3806
**Traps**	**31 May 2019**		
	**20 Year-Old Olive Grove**	**Almond Orchards**	**Cherry Orchard**
Yellow	13.92 a	1.67 b	0.46 a
Yellow with a circle-pattern	10.25 a	3.04 ab	0.46 a
Yellow with a line-pattern	9.63 a	4.04 a	0.29 a
F-value	1.21	4.25	0.25
*p*-value	0.3093	0.0202	0.7830

**Table 3 insects-14-00777-t003:** Mean capture ± SE of *P. spumarius* on the yellow sticky traps in the olive groves, and on the almond, and cherry trees. Values with the same letter are not significantly different.

Host Plant	May	June	July	August	September	October
Site A—2018						
Olive trees	-	-	-	2.0 ± 0.23 b	1.7 ± 0.18 b	1.9 ± 0.25 b
Almond trees	-	-	-	19.0 ± 1.78 a	20.4 ± 1.18 a	4.8 ± 0.63 a
F-value				411.27	538.18	53.65
*p*-value				<0.00001	<0.00001	<0.00001
Site A—2019						
Olive trees	0.9 ± 0.19 b	4.2 ± 0.58 b	1.4 ± 0.22 b	1.4 ± 0.27 b	1.1 ± 0.37 b	0.6 ± 0.15 b
Almond trees	2.9 ± 0.39 a	7.5 ± 0.63 a	2.5 ± 0.28 a	2.5 ± 0.30 a	3.5 ± 0.45 a	1.7 ± 0.23 a
Cherry trees	0.8 ± 0.26 b	2.2 ± 0.53 c	1.0 ± 0.25 b	1.2 ± 0.36 b	1.5 ± 0.42 b	1.4 ± 0.33 a
F-value	42.45	49.23	19.43	11.92	10.63	11.05
*p*-value	<0.00001	<0.00001	<0.00001	<0.00001	<0.00001	<0.00001
Site B—2019						
Olive trees	0.04 ± 0.05 a	1.4 ± 0.38 a	2.7 ± 0.63 a	2.1 ± 0.65 a	3.8 ± 0.83 a	0.9 ± 0.35 b
Almond trees	0.08 ± 0.08 a	1.8 ± 0.43 a	1.3 ± 0.42 b	1.0 ± 0.29 b	1.9 ± 0.36 b	1.7 ± 0.54 ab
Cherry trees	0.12 ± 0.09 a	1.9 ± 0.52 a	1.8 ± 0.42 ab	1.3 ± 0.35 ab	2.3 ± 0.54 b	2.3 ± 0.70 a
F-value	0.53	0.79	5.42	3.64	5.51	5.79
*p*-value	0.59	0.46	0.0071	0.0329	0.0066	0.0052

## Data Availability

The data presented in this study are available in this article and in the supplementary material.

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
