# Peer review of "Comparing Different Sticky Traps to Monitor the Occurrence of *Philaenus spumarius* and *Neophilaenus campestris*, Vectors of *Xylella fastidiosa,* in Different Crops"

_insects, 2023, doi:10.3390/insects14090777_

Round 1

Reviewer 1 Report

The manuscript is very well written and presents alternatives for monitoring insect vectors of the phytopathogenic bacterium Xylella fastidiosa. While it is widely known that yellow sticky traps primarily attract Hemiptera, I think a comparative study is very useful. In addition to the combined use with sweep nets to achieve comprehensive monitoring of target insects.

On the pdf I made some writing suggestions only.

Author Response

Thank you very much for your comments.

Point 1: Line 14, add “et al”.

Done

Point 2: line 24, add “et al”.

Done

Point 3: line 52 use italic

Done

Point 4: line 57, add Cercopidae and Membracidae too!

Done

Point 5: line 66, delete “and”

Done

Point 6: line 130, change “paragraph” with “section”

Done

Point 7: line 158, change “10” with “Ten”

Done

Point 8: line 159, change “4” with “four”

Done

Point 9: line 163, change “9” with “nine”

Done

Point 10: line 169, add “.”

Done

Point 11: line 176, add “)”

Done

Point 12: line 204, use italic

Done

Point 13: line 312, add “s”

Done

Point 14: line 318, delate “s”

Done

Reviewer 2 Report

Dongiovanni et al. Report the results of a two-year comparative study on the evaluation of yellow sticky traps and sweep net sampling for estimating spittlebug populations in Italy. This is an issue of interest as these insects transmit the plant pathogen Xylella fastidiosa. The study would be of interest to readers of Insects.
However, the manuscript requires major improvement before it would be suitable for publication.
Main points.
The clarity of the text needs to be improved throughout the manuscript. I have written suggestions and numerous numbered points on the manuscript.
The English needs editing.
It appeared to me that the same data were presented in Figures and Tables, which is unnecessary. Please let me know if this was not the case. I suspected that much of this material could be moved to supplementary material.
Numbered points written on scanned manuscript.
1.    Slightly more detail required on the patterns/design of yellow sticky traps in the Abstract.
2.    Unless you knew the densities a priori, suggest you delete the word “known”.
3.    Please provide information on the mean, (min and max) distance between traps.
4.    As samples were taken at different intervals, I wondered if the data should be converted to insects/trap/day to allow more accurate comparisons over time?
5.    Please provide an image of the traps and the circle and line designs. This could be included as a suppl. figure.
6a. What does consociated mean? Not an English word I have ever heard (as a native speaker).
6b. Please indicate the approximate size of the orchards (hectares).
7.    Insert “insecticide” before the word applications.
8.    Were traps of different designs rotated at each sample time to avoid position effects? If not, why?
9.    A decade is a period of 10 years in English (needs to be corrected throughout the manuscript).
10.    Did these different sampling intervals have an effect on the statistical analysis over time? Or was that avoided by analyzing each year separately. Please clarify here.
11.    Were sweep net samples taken at the same height as section 2.3?
12.    I did not understand random sampling units. Please clarify here.
13.    Area of orchards (hectares)?
14.    Please describe this protocol in a little more detail for immature insects.
15.    Were the data tested for normality and equality of variances prior to ANOVA?
16.    Were the data tested for normality and equal variances prior to t-tests?
17.    Please provide degrees of freedom (treatment and error) for F statistics.
18.    Secular means “non-religious” in English. Correct this in all Tables please.
19.    What do asterisks indicate in Fig 3?
20.    No issue here (please ignore this point)
21.    Please provide df values for F stats.
22.    Ditto.
23.    You did not evaluate attraction - - you counted TRAPPED insects, correct?
24.    Are the values in Table 2, Table 3, Table 4 Table 5 all means?
25.    Please provide df values.
26.    Where does the density estimate /0.25 m2 come from? I did not see sampling units of 0.25 m2 mentioned anywhere. The sticky traps appear to have a different area (ca. 600 cm2?).
27.    Also, is this really an insect density estimate or a mean trap capture value?
28.    Please indicate the meaning of asterisks in Figure 4, and 5.
29.    I could not read the text in Figure 6 (much too small).
30.    Please provide details of d.f. values for t-tests in a footnote to Table Suppl. S7, S8, S9
31.    Please remove “dec” from figure 7.
32.    I would argue that colors do not “drive” insect behavior. Factors such as hunger, reproduction, dispersal drive insect behavior which may be triggered or stimulated by certain colors.
33.    Recovery from what? Unclear.
34.    Do you mean habitat? Better just to say, “canopy vegetation”.
35.    Recovery again? Meaning? Do you mean shelter?
36.    I did not understand this phrase, please reword. Demarcated areas – reword to quarantine areas?

Needs editing.

Author Response

Thank you for your constructive comments.

Comments:

  1. Slightly more detail required on the patterns/design of yellow sticky traps in the Abstract.

Response: we have added more details on yellow sticky traps in the Abstract

  1. Unless you knew the densities a priori, suggest you delete the word “known”.

Response: we would prefer to leave the sentence unchanged, as the population density of juvenile stages has been evaluated in order to be able to set up the capture efficiency of yellow traps against adult individuals in different population density conditions

  1. Please provide information on the mean, (min and max) distance between traps.

Response: Traps distance followed the distance along the rows. This information was improved into the text.

  1. As samples were taken at different intervals, I wondered if the data should be converted to insects/trap/day to allow more accurate comparisons over time?

Response: The goal is not to obtain a comparison of the trapping capacity over time, but a direct comparison between the different kind of traps. The conversion would not change the result achieved

  1. Please provide an image of the traps and the circle and line designs. This could be included as a suppl. figure.

Response: Done

6a. What does consociated mean? Not an English word I have ever heard (as a native speaker).

Response: it is a traditional method of cultivation typical of the Mediterranean area, still quite widespread, with the simultaneous presence of different plant species within the same field. Specifically, in site B there were simultaneously presence of olive, almond and cherry trees, casually distributed in the orchard.

6b. Please indicate the approximate size of the orchards(hectares).

Response: we have added this information in the article

  1. Insert “insecticide” before the word applications.

Response: Done

  1. Were traps of different designs rotated at each sample time to avoid position effects? If not, why?

Response: to avoid position effects traps were rotated along each rows, and randomized according to the colour in a Latin Square Design

  1. A decade is a period of 10 years in English (needs to be corrected throughout the manuscript).

Response: Modified in all text

  1. Did these different sampling intervals have an effect on the statistical analysis over time? Or was that avoided by analyzing each year separately. Please clarify here.

Response: The different survey times did not affect the statistical analyses, as no comparisons were made over time, but only comparisons between the different kinds of traps for each sampling period, inspection site and year

  1. Were sweep net samples taken at the same height as section 2.3?

Response: yes, the sample have taken at the same height as previously section on olive canopies

  1. I did not understand random sampling units. Please clarify here.

Response: Simultaneously with each inspection of traps a pre-fixed number of sweeps were carried out. More specifically, were sampled 20 olive, almond or cherry trees using 10 sweeps for each plant spread around the entire canopy; in the ground vegetation were sampled 30 randomly distributed sampling units using 4 sweeps for each unit.

  1. Area of orchards (hectares)?

Response: we have added this information in the manuscript

  1. Please describe this protocol in a little more detail for immature insects.

Response: modified in the text

  1. Were the data tested for normality and equality of variances prior to ANOVA?

Response: Yes, before ANOVA test, we have verified the normality distribution and tested the equality of variance

  1. Were the data tested for normality and equal variances prior to t-tests?

Response: yes, according to the t-tests the data of both samples are prior normally distributed and used samples that have approximately the same variance

  1. Please provide degrees of freedom (treatment and error) for F statistics.

Response: Done

  1. Secular means “non-religious” in English. Correct this in all Tables please.

Response: Modified

  1. What do asterisks indicate in Fig 3?

Response: Asterisks indicated statistically significatively

  1. No issue here (please ignore this point)

Response: It is not clear the comment.

  1. Please provide df values for F stats.

Response: Done

  1. Ditto (df?)

Response: Done

  1. You did not evaluate attraction - - you counted TRAPPED insects, correct?

Response: Yes it is correct

  1. Are the values in Table 2, Table 3, Table 4 Table 5 all means?

Response: yes

  1. Please provide df values.

Response: Done

  1. Where does the density estimate /0.25 m2 come from? I did not see sampling units of 0.25 m2 mentioned anywhere. The sticky traps appear to have a different area (ca. 600 cm2?).

Response: The density population were determined on juvenile stages as reported at point 14

  1. Also, is this really an insect density estimate or a mean trap capture value?

Response: no, were determined considered the juvenile density population

  1. Please indicate the meaning of asterisks in Figure 4, and 5.

Response: the asterisks indicated the statistically significantly difference among the thesis

  1. I could not read the text in Figure 6 (much too small).

Response: modified

  1. Please provide details of d.f. values for t-tests in a footnote to Table Suppl. S7, S8, S9

Response: in all cases df = 19, generally in a T-test the value of df is not reported

  1. Please remove “dec” from figure 7.

Response: Done

  1. I would argue that colors do not “drive” insect behavior. Factors such as hunger, reproduction, dispersal drive insect behavior which may be triggered or stimulated by certain colors.

Response: I understand, I think the use of 'drive' is not very correct. Certainly colours influence certain behaviours, so I think the verb “influence” is more correct than drive

  1. Recovery from what? Unclear.

Response: we wanted to indicate: “….plants to use as shelters…”. “Recovery” is a mistake. We have modified the sentence

  1. Do you mean habitat? Better just to say, “canopy vegetation”.

Response: yes, means characteristics of canopy vegetation

  1. Recovery again? Meaning? Do you mean shelter?

Response: yes we would said shelter

  1. I did not understand this phrase, please reword. Demarcated areas – reword to quarantine areas?

Response: we have removed demarcated areas

In the manuscript:

  1. Line 15: italic

Done

  1. Line 17: change frequently with “most”

We prefer use “most” because in Europe is the main method to collects spittlebug adult vectors of Xf

  1. Line 18: change “paper” with “study”

Done

  1. Line 24: add “the”

Done

  1. Line 27: delete “s”

Done

  1. Line 29: add “was”

Done

  1. Line 30: see comment 1

See response of comment 1

  1. Lines 31-32: add “These” and delete “The results indicated that two”

Done

  1. Line 32: delete “with both” and add “and were”

Done

  1. Line 35: add “-s”

Done

  1. Line 37: delete “the”

Done

  1. Lines 43-44: delete “research investigation” and add “attention”

Done

  1. Line 46: delete “implied” and add “resulted in”

Done

  1. Line 47: delete “likewise” and add “such as”

Done

  1. Line 99: delete “as mean”

Done

  1. Line 101: delete “known”, see comment 2

See response of comment 2

  1. Lines 116-118: see comment 3

See response of comment 3

  1. Line: 119: see comment 4

See response of comment 4

  1. Line 127: add figure of traps, see comment 5

See response of comment 5

  1. Lines 131-136: see comments 6a and 6b

See response of comment 6a and 6b

  1. Line 132: “application”, see comment 7

See response of comment 7

  1. Line 135: “consociated?”

See response of comment 6a

  1. Lines 137-138: see comment 3

See response of comment 3

  1. Lines 141-142: see comment 8

See response of comment 8

  1. Line 144: “decade”?, see comment 9

See response of comment 9

  1. Lines 144-146: see comment 10

See response of comment 10

  1. Lines 149-152: see comment 3

See response of comment 3

  1. Line 154: delete “-s”

Done

  1. Line 156: delete “-ing”

Done

  1. Lines 157-160: see comment 11

See response of comment 11

  1. Line 159: see comment 12

See response of comment 12

  1. Line 164: ha? See comment 13

See response of comment 13

  1. Line 168: see comment 14

See response of comment 14

  1. Line 177: delete “every” and add “each”

Done

  1. Lines 179-181: see comment 15

See response of comment 15

  1. Line 185: add “means” and see comment 16

Done. See response of comment 16

  1. Line 189: delete: “α-level”

Done

  1. Line 204: italic

Done

  1. Line 207: delete “-s”

Done

  1. Lines 228-233: see comment 9

See response of comment 9

  1. Lines 235-236: see comment 17

See response of comment 17

  1. Line 237: add A) and B) in figure 1

Done

  1. Lines 244: add A) and B) in figure 1

Done

  1. Line 250: see comment 18

See response of comment 18

  1. Line 272: see comment 19

See response of comment 19

  1. Lines 301-308: see comment 20

See response of comment 20

  1. Lines 307-308: see comment 21

 See response of comment 21

  1. Line 318: delete “amount” and add “number”

Done

  1. Line 322: delete "-‘s”

Done

  1. Lines 322-327: add “df”, see comment 22?

Done

  1. Line 329: see comment 23

Delete “attraction on” and add “caught”

  1. Line 331

Table 2 has been correct

  1. Line 332: see comment 23

Delete “attraction on” and add “caught”

  1. Line 334

Table 3 has been correct

  1. Line 335: see comment 23

Delete “attraction on” and add “caught”

  1. Line 345: see comment 23

Delete “attraction ” and add “caught”

  1. Line 352: delete “the”

Done

  1. Lines 356-362: see comment 25

See response of comment 25

  1. Line 376: see comment 26

See response of comment 26

  1. Line 377: see comment 27

See response of comment 27

  1. Line 384: add “nymphs/0.25 m2

Done

  1. Line 386: delete “-ing”

Done

  1. Lines 387-394: see comment 25

See response of comment 25

  1. Line 394: “decade”?

It has been replaced

  1. Line 394: delete “-s” and add “-d”

Done

  1. Line 395: add “significantly”?

Done

  1. Lines 397-414

Add “df”

  1. Line 397: “decade”?

It has been replaced

  1. Line 402: delete “were” and add “was”

Done

  1. Line 402: delete "-ing”

Done

  1. Line 407: “decade”?

It has been replaced

  1. Line 411: delete “-ing”

Done

  1. Line 411: add “-s”

Done

  1. Line 413: delete "on” and add “in”

Done

  1. Line 414: delete "among” and add “between the”

Done

  1. Line 414: delete "was recorded”

Done

  1. Line 415: see comment 28

See response of comment 28

  1. Line 417: put “tree” before “canopy”

Done

  1. Line 421: see comment 28

See response of comment 28

  1. Line 425: see comment 29

See response of comment 29

  1. Line 426: figure 6

Text has been corrected

  1. Lines 429-450: see comment 30

See response of comment 30

  1. Line 429: supplementary Table 7

 Text has been corrected

  1. Line 437: put “adult” after “ spumarius

Done

  1. Line 448: put “adult” after “ spumarius

Done

  1. Line 477: delete “blu” and add “blue”

Done

  1. Lines 475-477: see comment 31

See response of comment 31

  1. Line 479: see comment 32

See response of comment 32

  1. Line 482: delete “have set up an intense program” and add “performed a detailed study”

Done

  1. Line 488: Xf

italic

  1. Line 494: add “was” and delete “-ly”

Done

  1. Line 495: delete “scarcely” and add “not”

Done

  1. Line 496: delete "-s”

Done

  1. Line 498: add “-s”

Done

  1. Line 500: delete " Whereas” and add “Sticky”

Done

  1. Line 502: delete "-’s” in  adult and add "-s” in population

Done

  1. Line 506: “habitus”?, see comment 34

See response of comment 34

  1. Line 520: delete “on” and add “to”

Done

  1. Line 526: see comment 33

See response of comment 33

  1. Line 531: add “to” and delete “-s”

Done

  1. Line 532: “insistent”??

Deleted  

  1. Line 532: “recover”, see comment 35

 See response of comment 35

  1. Lines 533-534: see comment 36

See response of comment 36

  1. Line 538: see comment 36

The phrase has been modified

  1. Line 541: delete “-s”

Done

  1. Line 542: delete “-s”

Done

  1. Line 544: add “a”

Done

  1. Line 553: add “in” and “,”

Done  

  1. Line 554: delete “can” and add “ may”

Done

  1. Line 559: “Xf”

italic

  1. Line 572: “:”

“,”

Reviewer 3 Report

With this study the authors aim to provide evidence supporting the use of a multi-sampling method approach for the  accurate estimation of presence and abundance of spittlebugs in crops potentially affected by Xylella fastidiosa.
Although, the purpose is relevant I found several points the must be improved and that I summarize here:

-title does not describe the study properly

-Introduction is well structured but it does not have an explicit statement for the hypothesis and the goals are poorly described

-Material and Methods: Subtitles poorly defined. The Section suffer of a lack of clear structure. There are two different sampling design for two years, why the author divided the description in 2 subheading is unclear. The sampling design lack of soundness in particular several choices need to be justified

-Results: Overall, the results section is very busy and a clear narrative is missing. The subheadings barely match those for the M&M. It is very difficult to understand what exactly the multiple testings are driving the reader. In particular, despite the very simple statement for the goal of the paper, the M&M and Results sections are difficult to collect.

-Discussion: I expected a much longer Discussion considering the length of the Results. I carefully check on the citations used and I think that the authors missed to cite several pieces of literature that are pertinent for this paper and results.
Several sentences do belong to Results section.
Some paragraph need to be better discussed.

Please find detailed feedback in the document attached below

The punctuation and grammar must be improved.

Author Response

Thank you very much for your comments.

-title does not describe the study properly

Response: taking into account the suggestion, we slightly modified the title, we hope it suites the reviewer request. 

-Introduction is well structured but it does not have an explicit statement for the hypothesis and the goals are poorly described

Response: we have improved the last part of the introduction section, making more explicit the aim of the work.

-Material and Methods: Subtitles poorly defined. The Section suffer of a lack of clear structure. There are two different sampling design for two years, why the author divided the description in 2 subheading is unclear. The sampling design lack of soundness in particular several choices need to be justified

Response: We added an initial paragraph to better clarify the experimental workflow, and as suggested by the reviewer we improved the structure of the M&M.

-Results: Overall, the results section is very busy and a clear narrative is missing. The subheadings barely match those for the M&M. It is very difficult to understand what exactly the multiple testings are driving the reader. In particular, despite the very simple statement for the goal of the paper, the M&M and Results sections are difficult to collect.

Response: We added more explanatory text in the results and moved to the supplementary file several tables so that the text can be more easy to ready. We modified the subheadings so that there is a march between M&M and results sections.

-Discussion: I expected a much longer Discussion considering the length of the Results. I carefully check on the citations used and I think that the authors missed to cite several pieces of literature that are pertinent for this paper and results. Several sentences do belong to Results section. Some paragraph need to be better discussed.

Response: We have modified some paragraphs and added others pertinent citations

In the manuscript:

  1. Line 2-3: Title needs to be reconsidered. if the occurrence is the parameter being estimated need to be declared in the paper

Response: Title improved.

  1. Line 30-31: Please add the authorship like you did for the bacterium. Also add (Hemiptera: Aphrophoridae)

Response: done

  1. Line 42: Summary:- English grammar revision necessary - Introduction is well structured but it does not have an explicit statement for the hypothesis and the goals are poorly described

Response: A paragraph has been added at the end of the introduction section to better clarify the goal of the paper. 

  1. Line 48-49: Switzerland is not listed. I would suggested to include it even if it is not a EU country.

Response: In this case, we wanted to indicate the countries where outbreak of Xylella have been reported  thanks to the surveillance programmes implemented by the EU. In Switzerland, fortunately no outbreaks have been so far reported.

  1. Line 54: the acronym is always italics as defined above

Response: done in all text

  1. Line 57: italics?

Response: yes, it was correct

  1. Line 62: first mention, the name should be spelled out

Response: done

  1. Line 73: second mention should be N.

Response: done

  1. Line 96-98: This sounds like an hypothesis, if so the authors should explicitly state their hypothesis and better explain its justification. If this is not an hypothesis then it is better to leave this out or move in discussion section.

Response: The sentence has been revised

  1. Line 99: This comparison is meant to estimate parameters of the populations only? How this is related to the surveillance programs? Does this study mean to get results that may be implemented in the field? Please specify.

Response: We have now modified this sentence and made clear the objective of the work.

  1. Line 100: which parameter(s) of the population this study is going to measure? if this is an estimation you should specify abundance? structure? distribution?

Response: We have now modified this sentence and made clear the objective of the work.

  1. Line 104: Subtitles poorly defined. The Section suffer of a lack of clear structure. There are two different sampling design for two years, why the author divided the description in 2 subheading is unclear. The sampling design lack of soundness in particular several choices need to be justified.

Response: We improved the structure.

  1. Line 105: This is not exactly a parameter of population, it should be specified in the last paragraph of the introduction

Response: Yes, it has been added.

  1. Line 106: How the authors can assess the color preference in the field? This kind of experiment are carried out with specific lab settings, and eventually compared with field results.

Response: We modified the sentence and pointed out that we evaluated the attractiveness of the traps.

  1. Line 115: Do they have different color reflectance? If so please specify, if not justify why the pattern should play a role in color preference assessment.

Response: we had information about the different efficiency of these traps for other classes of insects, thus we thought it would be worth to test them for spittlebugs.

  1. Line 116-118: How the authors can exclude confounding factors? It is likely that trapping may be driven by others component rather than color preference.

Response: Given that we used a randomized scheme, we believe that the differences recorded can be associated to the color.

  1. Line 119: Did the author considered the loss of stickiness on the trap over the time? two weeks is a long time, especially in certain climate conditions.

Response: the objective was the direct comparison between the different kids of color traps, positioned in the field at the same time; the possible loss of stickiness of the traps cannot influence the results achieved

  1. Line 124: "comparing the sticky traps" or "comparing the captures from different sticky traps"?

Response: Yes, improved.

  1. Line 125-126: This sentence is pretty confusing. I suppose the data gathered in the previous setting of the experimental design was meant to carried out some sort of evaluation and analysis. Please improve the meaning of this sentence.

Response: The sentence has been improved

  1. Line 126-128: As noted above please justify in the first subheading

Response: Yes improved

  1. Line 143-146: It is unclear the meaning of this subheading title, indeed the first subheading should also include some sort of comparison. Please improve.

Response: Yes improved

  1. Line 148: why we need an entire subheading for this? This is only a factor to test the variance of the response variable and should be included in a single subheading with the description of the experimental setting.

Response: Yes merged.

  1. Line 156: same comment as for 2.3

Response: Yes merged

  1. Line 170: change “was” with “were”

Response: It has been modified.

  1. Line 175: in results?

Response: Modified 3.1 and 3.2 with 2.1 and 2.2

  1. Line 177-178: why? and if so then the year is not in the model because there is no variation to measure

Response: Modified: data were analysed separately each month or period of observation

  1. Line 178: What do you mean? significant results?

Response: When with MANOVA test we have obtained statistically significant differences, data were individually analysed in ANOVA test

  1. Line 185: unclear. please improve

Response: ln the event that the results of the analysis of variance had provided significant results, the differences between the treatments were verified using Tukey tests

  1. Line 191: Overall, the results section is very busy and a clear narrative is missing. The subheadings barely match those for the M&M. It is very difficult to understand what exactly the multiple testings are driving the reader. In particular, despite the very simple statement for the goal of the paper, the M&M and Results sections are difficult to collect.

Response: As indicated above we improved the section.

  1. Line 210: the results support the need to better justify this assessment. Reflectance? other factors?

Response: we specified above that for other classes of insects results, the occurrence of black lines on the traps had an impact on the efficiency of the traps. 

  1. Line 293-294: “by different species belonging of the Issidae family.” how many?

Response: 3 species

  1. Line 295-300: How many species the authors used to run this analysis?

Response: all the species belonging to Cicadellidae and Issidae captured by sticky traps.

  1. Line 301-307: Did the author considered the amount of entomological glue if it differs among the traps, if not please specify.

Response: yes, we have considered these aspects, the traps had produced by a single company, did not differ in the amount of glue present on each trap

  1. Line 354-355: Did the author compared the sweeps and the trap at different heights in different months?

Response: yes

  1. Line 373: I think that 9 site in the Apulia region cannot be considered a Large scale setting

Response: Yes, we changed the title of the paragraph

  1. Line 374-375: necessary?

Response: Yes, we removed the sentence.

  1. Line 377: the categorization is confusing. The category apply to the sites, so it should be low-population sites ...Please improve

Response: we left the original sentence, since the sites have low, medium and high population density

  1. Line 405 : is it for the site? Please clarify

Response: the sentence is referred at a pool of 6 olive grove characterized by low density populations

  1. Line 478: I expected a much longer Discussion considering the length of the Results. I carefully checked on the citations used and I think that the authors missed to cite several pieces of literature that are pertinent for this paper and results. Several sentences do belong to Results section. Some paragraph need to be better discussed.

Response: We have modified some paragraphs and added others pertinent citations

  1. Line 480-481: This is not entirely true. I invite the authors to look for literature missing here. “[43]” is evidenziato

Response: We added others citations

  1. Line 482-483: The author have to justify why the same experimental design was not set for both years 2018 and 2019.

Response: Preliminarily in 2018, comparative evaluations were carried out between the different types of colored traps and the different kinds of yellow traps, so following these preliminary observations, the plain yellow sticky traps that provided the most significant data were selected and used for subsequent evaluations, also in 2019.

  1. Line 485: “experimentally”. I would avoid this word here.

Response: Yes, removed

  1. Line 488: I suggest the authors to cite specific studies on trap performance as well

Response: followed the sentence we have added others pertinent citation

  1. Line 489-490: “response to color traps” is highlighted

Response: the comment is not clear

  1. Line 491: “[27] is highlighted

Response: the comment is not clear

  1. Line 492: justification for choosing different pattern is missing

Response: we provided now the explanation.

  1. Line 498: can be moved at the end of the paragraph. Moreover, the discussion should not describe results.

Response: we improved the sentence.

  1. Line 505: highlighted

Response: the comment is not clear

  1. Line 506-508: This is also part of a standardize sampling design. Operators should define to swept only a certain type of canopy unless the purpose is to estimate the difference of canopy density.

Response: this sentence aims to provide an explanation about the differences recorded among the three crop species when collecting insects through sweep net, indicating that most likely the structure of the almond trees does not allow to have an efficient insect collection by sweep net.

  1. Line 514-516: the authors should report some details form Morente et al to let the reader better understand what the comparison is all about.

Response: we have modified the sentence

  1. Line 517-518: This is a results should not be reported here again

Response: we improved the sentence

  1. Line 519-521: This is a pure result. Should be moved in Results section

Response: we believe this is part of the critical discussion of the results.

  1. Line 521-523: This is a result, as above

Response: see comment above.

  1. Line 525-526: Please develop the discussion using the cited literature

Response: we improved the sentence

  1. Line 527-529: Is a result, see above

Response: see comment above.

  1. Line 539-541: why more research are needed?

Response: As we stated, in other habitats the ecology of this species can be different and thus it cannot be excluded a priori that this species is not relevant for the epidemiology of Xf infections.

Round 2

Reviewer 2 Report

Errors remain.

1. L152. Please explain in the text what you mean by promiscuous trees.

2. L155 Please explain what you mean by consociated agriculture in the text. Or replace this word by the correct term in English.

3. Section 2.5 Information is missing on the procedures employed to test the normality and equality of variances of the data prior to ANOVA or t-test.

4. L257. 258, 348, 349, 353 - the error degrees of freedom for F statistics are missing (only treatment d.f. values are shown).

5. L330, 331 - df values missing for F statistics (treatment df and error df).

6. L410, df value missing.

7. Trap rotation to avoid positional effects has not been mentioned in the text (see Point 8 of my previous review).

Needs careful editing.

Author Response

  1. Please explain in the text what you mean by promiscuous trees.

Response: We have modified promiscuous with consociated and explained consociated. Consociate is an agronomic correct term to explain the simultaneous presence of different plant species in the same orchards

  1. L155 Please explain what you mean by consociated agriculture in the text. Or replace this word by the correct term in English.

Response: We have modified in the text

  1. Section 2.5 Information is missing on the procedures employed to test the normality and equality of variances of the data prior to ANOVA or t-test.

Response: We added this information

  1. 258, 348, 349, 353 - the error degrees of freedom for F statistics are missing (only treatment d.f. values are shown).

Response: We added data mixing

  1. L330, 331 - df values missing for F statistics (treatment df and error df).

Response: We added data mixing

  1. L410, df value missing.

Response: Added on the text

  1. Trap rotation to avoid positional effects has not been mentioned in the text (see Point 8 of my previous review).

Response: Added on the text

Reviewer 3 Report

I really appreciated the effort of the authors to address the concerns raised. However, I feel they did not properly covered all the point raised along the manuscript and more edits are needed to improve it.
They clarified the experimental workflow, but unfortunately the soundness of the sampling design need more attention. There are still several points that I raised in the text that need to be improved, including the English Grammar and several typos along the text.
Although, authors responded point-by-point several questions were not answered or addressed.
Specifically, the following paragraphs need more attentions: lines 48-53; 101-110; 113-122; 207; the match of result's subheadings and M&M's subheadings which is a consequence of a clear statement of the goals and steps of the experimental workflow; in the discussion there are several point open to clarify, I am still convinced that results should not be repeated in the discussion but reworded and used to actually discuss them.

In general I would encourage the authors as much as possible to incorporate their answers in the text.

Please find my feedback attached below.

Please revise English grammar.

Author Response

all spittlebugs are vectors of Xf?

Response: although all xylem-feeders can be potential vectors of the bacterium, we edited the title and made specific reference to the 2 species targeted in our study

if the authors want to list only the countries where outbreaks have been recorded, it should be more explicit here. In their sentence they are saying that there was an outbreak in southern Italy, as a consequence there were "mandatory EU surveillance programs" that aim to discover "unreported entrenched foci" and then the authors list countries where the mandatory programs were applied. I know tha such kind of program have been applied in Switzerland as well with the aim to discover "unreported entrenched foci" indeed this resulted in the discovery of a foci  central Switzerland and consequent  eradication. . If the authors want to list countries where outbreaks happened they need to clarify it.

Response: we agree with the reviewer, however as reported in the EPPO database the finding in Switzerland was more related to an interception of infected plants rather than to an outbreak. In this paragraph we aimed to report the main outbreaks occurring in European countries, where the bacterium has established itself in open field and vectors have a role in the persistence of the infections.

The authors did make an effort to address the concerns raised in the first round of the revision. However, there are still several points to amend:

- please check typos and English grammar

- The authors stated the goals, they did not provide an hypothesis, why they think a multisampling approach is needed? The authors provided results from literature in the previous paragraph, what the authors concluded? is this literature driving them to set the goal of the study? If so, this should be included as hypothesis for the present study.

In my opinion there are several reason why it may be hypothesized a multisampling approach would be more efficient for the specific pathosystem they are investigating.

I strongly suggest to improve.

Response: we tried to further improve the last part of the introduction, we emphasized that sticky traps are less laborious and costly than sweep net, thus integrating their use in the surveillance programs brings several advantages.

Moreover, It is not clear to me why the authors specify that additional tools for monitoring have to be provided in area "where control strategies must be implemented as consequence of the occurrence or the high risk of establishment of Xf." Proper monitoring strategies need to be implemented for a reliable risk assessment which is the prerequisite for an effective control strategy. What I am saying is that tools are needed for the risk assessment regardless whether the control strategies "must" be implemented or not.

Response: we agree that tools are needed for the risk assessment. However it should be considered that given the widespread occurrence of spittlebugs, the need to proceed with monitoring programs is mainly related to the need to provide guidelines for the control applications.

Line 106-109: this sentence is unclear. How it is related with the paragraph 101-105? If it is a pilot study, should it be described before line 101?.

Response: The whole paragraph was improved.

I do not understand the rationale of the experiment, as pointed out above: if one yellow stick trap pattern performs better in an orchard type (olive, cherry or almond) this is not a reason, from my point of view,  to focus on just one pattern for testing ground versus canopy capture efficiency.

This is particularly true in experiment in the field where multiple factors (including confounding ones) need to be controlled. there are very effective statistical methods that may be used to address this precific problem (e.g. Manova, adonis ...).

I suggest the authors to re-evaluate their data in this perspective.

Response: Actually as showed the main advantage of using the traps is for capturing insects on the canopies where sweep net is less efficient and more laborious, having this in mind we focus on assessing the most attractive traps hanged on canopies, however we also tested them on the ground vegetation showing they were more efficient than sweep net, thus indicating that still the yellow sticky traps were highly attractive even when they were placed on ground vegetation. 

Here is an example where the authors did not address my comment form the previous round of revision. The authors stated" Given that we used a randomized scheme, we believe that the differences recorded can be associated to the color." they should write it in the text and also it would be good if they provide citation from the literature that this assumption can be made or any evidence supporting the assumption.

Response: we believe that by indicating that the traps were randomized, should be enough to ensure that the randomization excluded that results were affected by the location of the traps.

Are these subheadings 2,1, 2.2 ...matching the steps that the authors defined in the introductory paragraph of Materials and Methods? If so make it clear. Moreover, what it means "different" how they are different?Here is an example where the authors did not address my comment form the previous round of revision. The authors stated" Given that we used a randomized scheme, we believe that the differences recorded can be associated to the color." they should write it in the text and also it would be good if they provide citation from the literature that this assumption can be made or any evidence supporting the assumption.

Response: we further improved the subtitle.

Is it linked to lines 118-119? if so make it clear

Response: we edited this first part of the paragraph

How this is related with the goals stated in the Introduction or the steps delineated in the first paragraph of the Material and Methods?

Please improve.

Response: we further improved this part

same as for the year, why it was analyzed separately?

Response: to gather information on the seasonal fluctuations we analyzed the data for each month

I do not understand the answer form the authors:"ln the event that the results of the analysis of variance had provided significant results, the differences between the treatments were verified using Tukey tests"

Why they did not check non-significant ANOVA with significant multiple pairwise comparisons?

Response: we agree with the reviewer, however as this analysis did not provided additional insights we did not included the data.

The answers to my comments in the first round of the revision should be incorporated in the text when possible. Here for example the answer from the authors may be incorporated in the M&M or discussion.

as above, comments need to be incorporated in the text, not just in the point-by-point response, in particular if it will improve the clarity and reproducibility of the experiment

Response: we are sorry but it is not clear whatelse we should add in this regard

The comment was double highlighted and was there I retype it for your convenience. The comment was:

However, the authors should also mention that the sweep net results strongly depends on the sampling design and differ with the operator enthusiasm to swept vegetation.

Response: we have now specifically added this aspect in the discussion section.

Round 3

Reviewer 3 Report

please check

line 125: typo

Also check English grammar to avoid misunderstanding, for example

lines 108, 112, 122 (do you mean both on the ground and on the canopy?), 185...

I think that English language may be improved.

Author Response

Dear reviewer,

I would like to thank you on behalf of myself and the other authors for your comments and suggestions, which enabled us to improve our manuscript. As you suggested, the manuscript was revised by an experienced native English speaker.

I hope that the English of the text has been improved as requested.

Yours sincerely

Vincenzo Cavalieri